# Whole Genome Analysis and Assessment of the Metabolic Potential of *Streptomyces carpaticus* SCPM-O-B-9993, a Promising Phytostimulant and Antiviral Agent

**DOI:** 10.3390/biology13060388

**Published:** 2024-05-28

**Authors:** Yulia Bataeva, Yanina Delegan, Alexander Bogun, Lidiya Shishkina, Lilit Grigoryan

**Affiliations:** 1State Research Center for Applied Microbiology and Biotechnology, 142279 Obolensk, Russia; kadnikova_lidiya@mail.ru; 2Institute of Biochemistry and Physiology of Microorganisms, Federal Research Center “Pushchino Scientific Center for Biological Research of the Russian Academy of Sciences” (FRC PSCBR RAS), 142290 Pushchino, Russia; y.delegan@yandex.ru (Y.D.); bogun62@mail.ru (A.B.); 3Department of Biology, Tatishchev Astrakhan State University, 414056 Astrakhan, Russia; lilyagrigoryan90@mail.ru

**Keywords:** *Streptomyces*, phytostimulation, genome analysis, secondary metabolites, plant protection

## Abstract

**Simple Summary:**

Natural microorganism-derived products are an essential source of valuable pharmaceuticals and agrichemicals. *Streptomyces* spp. are the most environmentally abundant bacteria that are capable of producing various valuable natural products exhibiting significant biological activity to be used in such industries as medicine, environmental science, food industry, and agriculture. However, many natural products among *Streptomyces* bacteria remain unstudied. Because of the rapid increase in antimicrobial and pesticide resistance, it is currently extremely topical to develop novel antibiotics, antiviral agents, agrichemicals, and other substances. The modern methods of molecular biology allow one to detect antimicrobial resistance gene clusters in the bacterial cell. Our study detected antibiotics and terpenes in cells of investigated bacteria; these compounds are of significant interest for the plant-growing sector.

**Abstract:**

This work aimed to study the genome organization and the metabolic potential of *Streptomyces carpaticus* strain SCPM-O-B-9993, a promising plant-protecting and plant-stimulating strain isolated from brown semi-desert soils with very high salinity. The strain genome contains a linear chromosome 5,968,715 bp long and has no plasmids. The genome contains 5331 coding sequences among which 2139 (40.1%) are functionally annotated. Biosynthetic gene clusters (BGCs) of secondary metabolites exhibiting antimicrobial properties (ohmyungsamycin, pellasoren, naringenin, and ansamycin) were identified in the genome. The most efficient period of SCPM-O-B-9993 strain cultivation was 72 h: during this period, the culture went from the exponential to the stationary growth phase as well as exhibited excellent phytostimulatory properties and antiviral activity against the cucumber mosaic virus in tomatoes under laboratory conditions. The *Streptomyces carpaticus* SCPM-OB-9993 strain is a biotechnologically promising producer of secondary metabolites exhibiting antiviral and phytostimulatory properties.

## 1. Introduction

Actinobacteria are characterized by high ecological plasticity, labile enzymatic systems, and a powerful and complex secondary metabolism [1,2].

According to the research into abundance and species diversity of *Actinobacteria*, *Streptomyces* spp. constitute 80–95% of all soil-dwelling actinobacteria. The overwhelming majority (70–80%) of all the known biologically active microbial secondary metabolites are produced by *Streptomyces* spp. [3].

The genus *Streptomyces* includes spore-forming, filamentous, and Gram-positive *Actinobacteria* [4,5]. Members of the genus *Streptomyces* are able to survive under adverse environmental conditions, while retaining metabolic activity for a long time, and degrade natural and synthetic substances as they possess enzymes with a wide substrate specificity. These bacteria produce a great variety of chemical components (polyketides, peptides, macrolides, indoles, aminoglycosides, terpenes, etc.) [6,7,8] through which they exert regulatory effects on the plant and control development of phytopathogens [9].

Members of the *Streptomyces* genus can affect phytopathogens either directly by producing antibiotics, siderophores, hydrolytic, or detoxifying enzymes, or indirectly by stimulating host plant growth through synthesis of phytohormones, increasing their resistance to diseases, forming mechanisms of induced and/or acquired system resistance, or simply by competing with phytopathogens for available nutrients [10,11].

Functions of the genes annotated in the genomes are currently poorly understood and require further comprehensive studies [12]. Using in silico genome analysis methods, Streptomyces genomes have been found to contain 25–70 biologically active compounds, but only a small fraction of these compounds are synthesized in the laboratory using culturing methods [13]. Modern sequencing techniques pose serious computational challenges because of short lengths of the sequenced fragments and large data volumes, which especially affect the functional annotation of the genomes of *Streptomyces* strains, since the latter contain many proteins with repeats, multiblock structures such as polyketide synthases, non-ribosomal peptide synthases (NRPSs), and serine threonine kinases [14,15]. The functional genes of *Streptomyces* are currently being intensively studied [16,17]. Thus, the genome of the *S. clavuligerus* strain contains many biosynthetic gene clusters (BGCs) of secondary metabolites such as staurosporine [18], moenomycin [19], terpenes, pentalenes, phytoenes, siderophores, and lantibiotics [20].

Given the high degree of polymorphism of *Streptomyces*, it is undoubtedly important, from a scientific and practical point of view, to study the level of specificity and biological activity of *S. carpaticus* strain SCPM-O-B-9993 [21]. The S. *carpaticus* SCPM-O-B-9993 strain was isolated from brown semi-desert soils with very high salinity in the Astrakhan Region of the Russian Federation. The laboratory studies conducted previously demonstrated that this strain exhibits phytostimulatory, antifungal, antioxidant, insecticidal, and antiviral activities, thus being of interest for the plant-growing industry [22]. This work aimed to investigate the genome organization and metabolic potential of *Streptomyces carpaticus* strain SCPM-O-B-9993, a promising plant-protecting and plant-stimulating strain.

## 2. Materials and Methods

### 2.1. Strain Cultivation

The *S. carpaticus* SCPM-O-B-9993 strain was cultured in starch casein agar at 28 °C. The composition of starch casein agar (g/L distilled water) was as follows: soluble starch, 10.0; casein, 0.3; KNO_3_, 2.0; K_2_HPO_4_, 2.0; MgSO_4_·7H_2_O, 0.05; NaCl, 2.0; CaCO_3_, 0.02; FeSO_4_·7H_2_O, 0.01, agar, 20.0. For the experiment, a 1 cm^2^ fragment of 7-day strain colonies with aerial and substrate mycelium was reinoculated from solid starch casein agar to 1 L of liquid growth medium of the same composition. The SCPM-O-B-9993 strain was cultured during 168 h at 28 °C under constant stirring in a shaker (80 rpm). During 7-day cultivation, an aliquot of the culture medium with cells (suspension) was used to inoculate solid starch casein agar (0.1 mL) every 24 h for calculating colony-forming units (CFU) and testing culture purity; cell count was estimated by measuring optical density of the suspension at 440 nm and used to determine phytostimulatory and antiviral activities. Strain cells were examined using an Amplival microscope (Zeiss, Oberkochen, Germany).

### 2.2. Phytotoxicity Assessment

Phytotoxicity of the SCPM-O-B-9993 strain suspension was studied by the wet cell method using seeds of Ducat cress (*Lepidium sativum*). The seeds were pre-sterilized in 70% ethanol during 3 min and then washed with distilled water several times. Each Petri dish was bottomed with a circle made of filter paper; 30 cress seeds were placed on it, and then 5 mL of the suspension was added (distilled water was used as a control). The dishes were exposed to 20 °C under natural light conditions for 3 days. The experiment was performed in two replicates. Germination was then calculated; root length and stem height of seedlings were measured using a ruler.

The results were analyzed using the conventional mathematical statistics methods in the Excel (2023) and Biostat (https://www.analystsoft.com/en/products/biostat/) software.

### 2.3. Studying Antiviral Activity

Tomato plants (*Solanum lycopersicum*) at the growth stage of 3–4 true leaves were infected with the cucumber mosaic virus (CMV) by placing the inoculum onto the upper surface of leaf laminae. A CMV isolate derived from open-ground tomato plants (F1 Adonis bred in Russia) was used as material for infection.

Seven days after treatment with the CMV inoculum, tomato plants were sprayed with a suspension of the strain. Control plants were sprayed with sterile distilled water (K^−^) and pharmaiodine (Scientific Production Center Farmbiomed, Moscow, Russia) (K^+^). An aqueous solution of pharmaiodine was prepared immediately before use: 3–5 g/10 L of water. Each experimental and control group consisted of ten plants. The experiment was performed in two replicates. In all the experiments, the plants were watered to the root zone with settled tap water in the morning. Throughout the entire experiment, the plants were subjected to two treatments with bacterial suspensions with a 7-day interval. Three days after the second spraying, immunochromatography tests were conducted to detect infection by the phytovirus using ImmunoStrip CMV (*Cucumber Mosaic Virus*) Test Kit Flashkits^®^ (Agdia Inc., Elkhart, IN, USA) consisting of a microtiter strip impregnated with an alkaline enzyme and bilaterally coated with anti-phytovirus antibodies and a buffer-filled bag for sample extraction. For diagnostics, a leaf portion (0.15 g) of each plant was placed into the lower part of the bag containing phosphate-buffered saline (0.2 M, pH 7.4), and a plant tissue sample was ground between mesh pads. An immunostrip was submerged into the resulting suspension until the mark “sample” and left in a vertical position for 30 min to further process the analysis results. Staining of the band whose mobility corresponded to the positive control suggested that plants had been infected. Plants infected by the virus were characterized by leaf deformation, a yellow-green mosaic pattern, and significant growth delay.

### 2.4. Genome Analysis

The strain genome was sequenced and completely assembled [22]. The genome was annotated with the NCBI Prokaryotic Genome Annotation Pipeline (PGAP) version 4.6 [23], Prokka [24], and RAST [25]. The whole-genome tree was built using the TYGS web service [https://tygs.dsmz.de/ (accessed on 1 May 2023)]. The genomic map was built using the Proksee.ca web service (accessed on 1 May 2023) [26]. The average nucleotide identity (ANI) value was calculated using the Ezbiocloud service (accessed on 1 May 2023) [27]; the *DNA*–*DNA* hybridization (DDH) value, using the Genome-to-Genome distance calculator (Formula (1) (i.e., GBDP formula *d*_0_)) [28]. Chromosome alignment was performed using Mauve software, ver. 2.4.0 (21 December 2014) [29]. Pangenome analysis and searching for unique genes were performed using OrthoVenn service (https://orthovenn3.bioinfotoolkits.net/, accessed on 28 February 2024) [30]. Searching for gene clusters responsible for secondary metabolite production was conducted using the AntiSmash service (https://antismash.secondarymetabolites.org/, accessed on 28 February 2024) [31]. Functional analysis based on metabolic and regulatory pathways was carried out using KEGG (https://www.kegg.jp/blastkoala/, accessed on 28 February 2024) [32]. SNP searches were performed using Nucmer from the Mummer v. 4.0.0 package [33].

## 3. Results

### 3.1. Morphological Features of the Strain

The *S. carpaticus* SCPM-O-B-9993 strain has dark brown aerial mycelium and cherry-red substrate mycelium (Appendix A). The optimal growth temperature is 28 °C; the optimal pH is 7.0–7.1 [22]. Spores are straight or twisted, and short. Spores are oval or globular with a dense shell, sized 0.5–1.0 × 1.0–1.1 µm.

### 3.2. Evaluation of Productivity as well as the Phytostimulatory and Antiviral Properties of the Strain

During the first 12 h of growth, the strain is in the lag phase when the cells are adapting to the nutrient medium. The cells and mycelium enter the exponential growth phase by 12–24 h. At this time, strain SCPM-O-B-9993 actively produces metabolites, their cells multiply, and their numbers increase to 2.0 × 10^9^ CFU/mL. Starting with 48 h of exposure, the culture begins to enter the stationary growth phase and remains in it for five days. Abundance does not decrease even on day 7. The optical density changes in accordance with strain biomass growth (Figure 1).

When studying the effect of strain SCPM-O-B-9993 suspension on cress seeds, a toxic effect on germination of 4-day and 7-day cultures and positive control pharmaiodine was detected (Appendix A). A toxic effect was observed in germination of less than 30% of seeds. The concentration of cells, spores, mycelium, and the number of metabolites in the suspension cultivated under laboratory conditions in liquid nutrient medium increased with cultivation duration, which increased the toxicity of the suspension. However, it should be taken into account that bacteria entering natural conditions in plant treatments, at a concentration of 10^9^ CFU/mL, are distributed in soil or on plants in lower quantities than when they are under laboratory flask conditions due to lack of substrate, environmental factors, etc. In addition, when microorganisms enter the soil, a microbial pool is formed, which further leaves the number of cells no more than 10^4^ CFU/mL.

The longest root (23.89 mm) was found for treatment with a 3-day suspension of the strain. The cultivation period of the strain influenced cress growth; 1-, 2-, 6-, and 7-day cultures exhibited an inhibitory effect on root and shoot growth. It is quite natural that 6- and 7-day-old cultures inhibited plant growth because of the accumulation of metabolites at this stage.

The data indicate that *S. carpaticus* strain SCPM-O-B-9993 exhibits an antiviral activity against CMV on tomatoes under laboratory conditions relative to positive and negative controls (Table 1, Appendix A). When comparing the antagonistic properties of the strain suspension by hours of cultivation, it was found that 3-day suspensions of the strain exhibited the maximum antiviral activity: all the treated plants were symptom-free of the virus. Hence, 3-day cultivation of the strain is optimal, since the suspension showed effective phytostimulatory properties, while having an inhibitory effect on CMV. The study of secondary metabolites of the 3-day suspension of the strain showed the presence of alcohols, aldehydes, hydrocarbons, esters, sulfates, and other groups of low-molecular-weight organic compounds with high biological activity [21].

Suspensions of the strain after 24 and 48 h of cultivation were completely ineffective against the phytovirus. It was found that 4-day and 5-day suspensions of the strain continued to exhibit an antiviral activity, but it was insignificantly lower than that of 3-day suspensions. The antiviral activity of the 6- and 7-day suspensions of the strain was approximately 50%. Ara et al. [34] found out that symptoms of the tobacco mosaic virus in Datura plant treated with *Streptomyces* strains decreased after 7-day incubation due to the effect of bioactive metabolites of the strains.

### 3.3. Taxonomic Positioning of the Strain SCPM-O-B-9993

The genomes of *Streptomyces* strains include large unknown BGCs, making them a potential repository for discovering biotechnologically valuable compounds. Diverse molecules exhibiting an antibacterial activity encoded in the genome exist in the “silent” repressed state. Therefore, there arises a need for using genome editing and metagenomic analysis methods to identify new biosynthetic clusters of antibiotics and alter expression of the respective genes, which would potentially enable synthesis of novel molecules exhibiting an antibacterial activity [35].

Strain SCPM-O-B-9993 is the only representative of the *S. carpaticus* species whose genome has been completely sequenced [22]. The genome assembly consisted of a 5,968,715 bp long single linear chromosome with 72.84% GC composition (Figure 2). No plasmids were detected. During genome annotation and analysis, 5206 protein-coding sequences, 60 tRNA sequences, 15 rRNA sequences (5–5S, 5–16S, and 5–23S), and eight CRISPR loci were identified.

A BLAST search for housekeeping gene sequences (*gyr*B, *rpo*B, *trp*B, *rec*A) was performed to determine its closest relatives among *Streptomyces* strains with completely sequenced genomes (Appendix A). *Streptomyces harbinensis* NA02264, *Streptomyces xiamenensis* 318 and *Streptomyces* sp. XC2026 were found to be most closely related to strain SCPM-O-B-9993 (Table 2).

Strains SCPM-O-B-9993 and NA02264 are deposited in the Genbank database as representatives of different species. However, the ANI and DDH values show that these strains are very likely to belong to the same species (Table 3). Hence, the threshold for species assignment by DDH is 70%, while strains SCPM-O-B-9993 and NA02264 have DDH values >90%.

The genome of the type strain *S. carpaticus* has not been sequenced yet (July 2023). However, the genome of the type strain *S. harbinensis* CGMCC4.7047T (FPAB000000000000.1) has been sequenced, which makes it possible to calculate the main parameters of species membership (Table 4).

Considering that the ANI and DDH values of strain SCPM-O-B-9993 compared to the type strain of *S. harbinensis* exceed the thresholds for species assignment (ANI > 95% and DDH > 70%), we assume that strain SCPM-O-B-9993 is probably a representative of the species *S. harbinensis*, but the final taxonomic position of the strain will be possible only after the genome sequence of the type strain of *S. carpaticus* appears in the Genbank database.

The phylogenetic tree (Figure 3) demonstrates that the strain SCPM-O-B-9993 is also close to *S. harbinensis* (*Streptomyces gingkonis* species is not validated, so we do not consider it).

Alignment of the genomes of *Streptomyces carpaticus* strain SCPM-O-B-9993 and *Streptomyces harbinensis* strain NA02264 (Figure 4) relative to each other shows that the main blocks (marked with one color) retain a similar arrangement on the chromosomes and in general, the structures of the genomes are very similar. Only a few displacements of gene blocks can be noted (marked with vertical bars).

In the *S. harbinensis* NA02264 genome relative to the *S. carpaticus* SCPM-O-B-9993 genome, 20,153 single nucleotide substitutions (SNPs) were found, which are evenly dispersed throughout the genome. Single nucleotide substitutions accounted for 0.34% of the total genome length.

### 3.4. Analysis of the Genome of Streptomyces carpaticus Strain SCPM-O-B-9993 and Its Closest Relatives

We analyzed the genome of the strains listed in Table 1 as the closest relatives of strain SCPM-O-B-9993 (Figure 5A), and separately, the pangenome of the pair *Streptomyces carpaticus* SCPM-O-B-9993 and *Streptomyces harbinensis* NA02264 (Figure 5B) as the strains closest to each other among the *Streptomyces* whose genomes were sequenced. It was of interest to identify whether these two strains, whose genomes are nearly identical in terms of structure, have genes that are unique relative to each other.

The genome of four *Streptomyces* strains is represented by 5700 orthologous clusters (OCs) (clusters of genes in different species that originated by vertical descent from a single gene in the last common ancestor), of which 4025 (70.6%) are core. Expectedly, the SCPM-O-B-9993/NA02264 pair had the maximum (of all pairs with strain SCPM-O-B-9993) number of OCs unique to the pair: 466 are characteristic only of this pair and absent in the other strains (Appendix A).

The genome of the *Streptomyces carpaticus* SCPM-O-B-9993 and *Streptomyces harbinensis* NA02264 pair is represented by 4851 clusters, of which 4827 (99.4%) are core. Despite the close relatedness, each strain has unique CDSs, but most of them cannot be classified at this time.

### 3.5. Functional Annotation of Streptomyces carpaticus Strain SCPM-O-B-9993

The genome of strain SCPM-O-B-9993 contains 5331 coding sequences, of which 2139 (40.1%) are functionally annotated (Figure 6).

Since the strain SCPM-O-B-9993 is a biotechnologically promising producer of secondary metabolites, the genetic organization of secondary metabolite production clusters was analyzed.

Antibiotic biosynthesis by microorganisms plays an adaptive role; it is an inherited feature of microorganisms and is regulated by specialized genes. 3-Amino-5-hydroxybenzoic acid (AHBA) is the starting unit in the biosynthesis of ansamycin antibiotics by *Streptomyces* bacteria [36]. The genome of strain SCPM-O-B-9993 contains a sequence of genes whose products control the pathway of AHBA biosynthesis from UDP glucose, consisting of ten reactions. The terpenoid biosynthesis pathway in the genome of strain SCPM-O-B-9993 involves production of key metabolites such as isopentenyl pyrophosphate, geranyl-PP, farnesyl-PP, and geranyl-geranyl-PP, which are precursors of many secondary metabolites in *Streptomyces*.

We found the following highly conserved secondary metabolite production clusters in the strain genome (>80% structural similarity to similar clusters in other strains) (Table 5).

The organization of the coelibactin gene cluster in strain SCPM-O-B-9993 is completely identical to that of *S. harbinensis* (Appendix A). Differences between them are present mainly at the level of single nucleotide substitutions (SNPs), except for a few extended regions (Appendix A).

The antimicrobial properties of the strain may be due to production of ohmyungsamycin, pellasoren, and naringenin. Ohmyungsamycins are cyclic peptides first isolated from marine *Streptomyces* [37,38]. They are synthesized by a non-ribosomal peptide synthetase. Kim et al. [39] reported their activity against *Mycobacterium tuberculosis* and cancer cells in humans, with OMS A showing greater activity than OMS B. Considering that strain SCPM-O-B-9993 possesses the ohmyungsamycin BGC, it can be considered promising in medicine as well. There are some structural differences among the ohmyungsamycin BGC in SPM-O-B-9993 and NA02264 strains. In the NA02264 genome, in this region there resides (there are no in SCPM-O-B-9993) IS110 family transposase and a region embedded between the hypothetical protein and α/β hydrolase, having a length of 8393 bp and containing several genes both annotated (tyrosine-type recombinase, NAD-dependent epimerase/dehydratase family protein) and hypothetical.

Previously, naringenin biosynthesis was believed to be characteristic only of plants [40,41]. Naringenin is a flavonoid whose biosynthesis has been repeatedly reported in citrus trees (lemons, oranges, etc) and tomatoes [42]. Álvarez-Álvarez et al. [43] first showed its biosynthesis in *Streptomyces clavuligerus*. Naringenin exhibits anti-inflammatory, chemoprotective, and antitumor properties [44,45]. The naringenin BGC is completely structurally similar to that of the strain *Streptomyces narbonensis* NA02264.

*Sorangium cellulosum* is the best-known producer of pellasoren [46]. However, the gene cluster for biosynthesis of this compound is found in the genomes of *Streptomyces* species, in particular in the type strain *S. harbinensis* (Appendix A). Nothing is known about the antimicrobial properties of pellasoren, this compound was reported to exhibit cytotoxic effects against cancer cells [47,48]. Pellasorene BGC in the SCPM-O-B-9993 and NA02264 strains have the following structural difference: the type I polyketide synthase gene is present between fatty acyl-*AMP* ligase and *SDR* family NAD(P)-dependent oxidoreductase genes in the SCPM-O-B-9993 strain, having coordinates 543.262–544.999 (1737 bp). The NA02264 strain does not contain it.

The detected BGCs in SCPM-O-B-9993 strain have a strong potential for being used in plant protection, since they exhibit antimicrobial properties.

Investigation of the component composition of suspension and extracts (water-alcohol, methanol, and hexane) of *S. carpaticus* strain SCPM-O-B-9993 showed the presence of secondary metabolites: alcohols, aldehydes, hydrocarbons, esters, sulfates, and other groups of low-molecular-weight organic compounds (LMCs). Alcohols and esters prevailed among LMCs for all the extraction variants. The identified LMCs have valuable properties from the agricultural point of view: antiviral, antimicrobial and antitumor (ethyl 5-(pyridin-4-yl)-1H-pyrazol-3-carboxylate); bactericidal, fungicidal, and antiseptic properties (1,2-hexanediol); and insectoacaricidal ones (isopropyl myristate). 1-Dodecanol is a component of pheromones, sex attractants and surfactants for controlling insect pests [21].

Annotation of the Streptomyces genomes shows that different BGCs are present. Thus, the genome of the *Streptomyces tendae* strain UTMC 3329 shows the presence of clusters of the genes encoding polyketides, ribosomally and non-ribosomally synthesized peptides. Various genes have been discovered in the xenobiotic degradation pathway and heavy metal resistance [49]. An analysis of *S. avermitilis* genome revealed the feasibility of polyketide synthesis [50]. Genome mining and manipulations with the genome, and with antibiotic BGC in particular, have a significant potential for identifying novel molecules exhibiting an antibacterial activity [51].

## 4. Conclusions

The studies revealed clusters of biosynthesis of secondary metabolites of the strain SCPM-O-B-9993: terpenoids and ansamycin antibiotics. Under laboratory conditions, the strain exhibited phytostimulatory and antiviral properties. The most effective cultivation period of the strain was 72 h, during which the culture went from exponential to stationary growth phase with the production of a spectrum of bioactive metabolites.

There are alternatives to genome analysis, such as metabolomic analysis (coupled to dereplication of known compounds) to assess the presence of potentially new compounds in any Streptomyces culture. The *Streptomyces carpaticus* SCPM-OB-9993 strain is a biotechnologically promising producer of secondary metabolites exhibiting antiviral and phytostimulatory properties.

## Figures and Tables

**Figure 1 biology-13-00388-f001:**
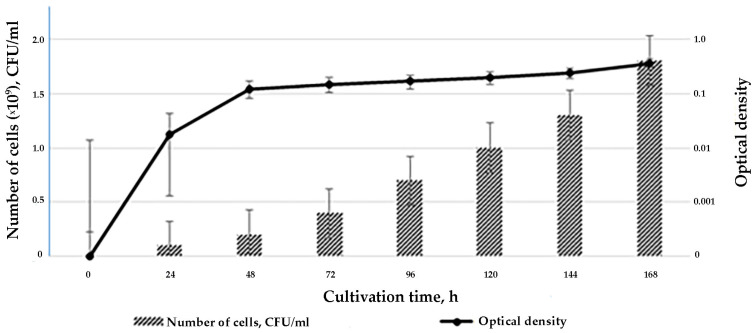
The growth curve of *S. carpaticus* strain SCPM-O-B-9993 when cultured on liquid starch–casein medium.

**Figure 2 biology-13-00388-f002:**
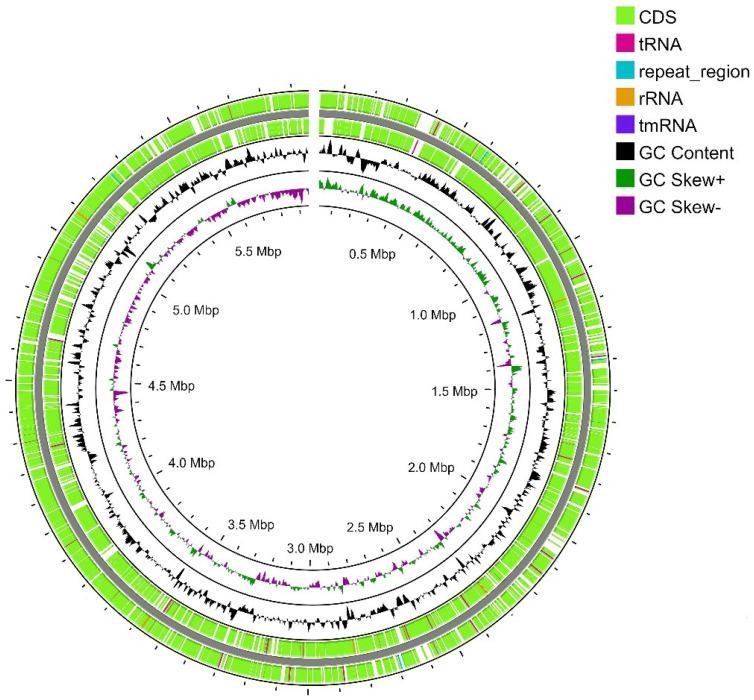
Map of the linear (gap on the top of the figure) chromosome of strain SCPM-O-B-9993. From outside to the center: all CDS and RNA genes on the forward strand, all CDS and RNA genes on the reverse strand, GC content, and GC skew.

**Figure 3 biology-13-00388-f003:**
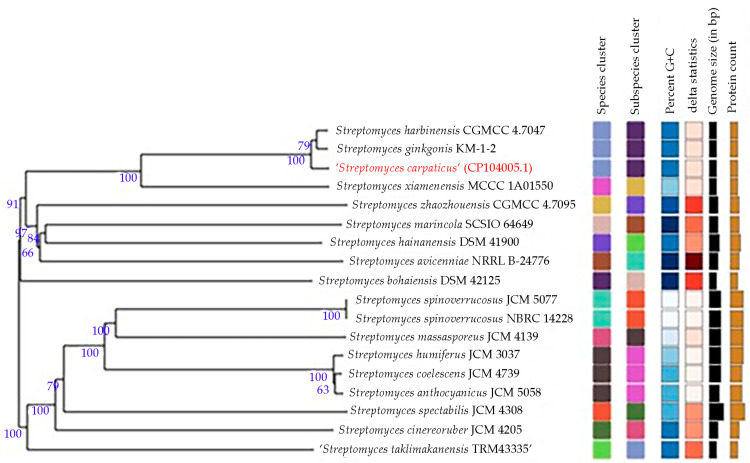
Whole-genome tree demonstrating the position of strain SCPM-O-B-9993 within the genus *Streptomyces*.

**Figure 4 biology-13-00388-f004:**
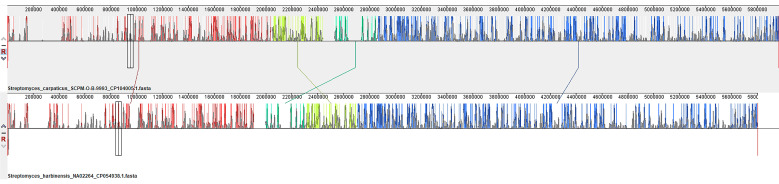
Mauve visualization of locally collinear blocks identified between chromosomes of *Streptomyces carpaticus* SCPM-O-B-9993 and *Streptomyces harbinensis* NA02264. Vertical bars demarcate interchromosomal boundaries.

**Figure 5 biology-13-00388-f005:**
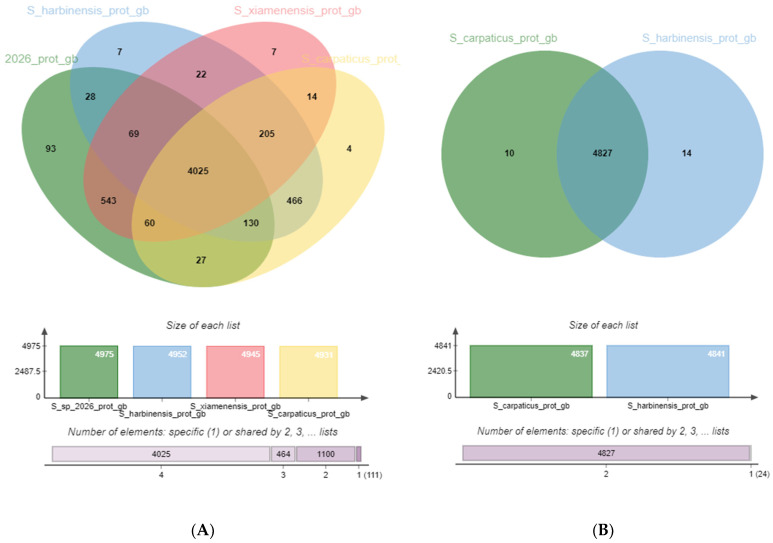
Venn diagram depicting the genome of *S. carpaticus* SCPM-O-B-9993 and its relatives: (**A**) *Streptomyces harbinensis* NA02264, *Streptomyces xiamenensis* 318, *Streptomyces* sp. XC2026; (**B**) *Streptomyces harbinensis* NA02264 as the closest relative.

**Figure 6 biology-13-00388-f006:**
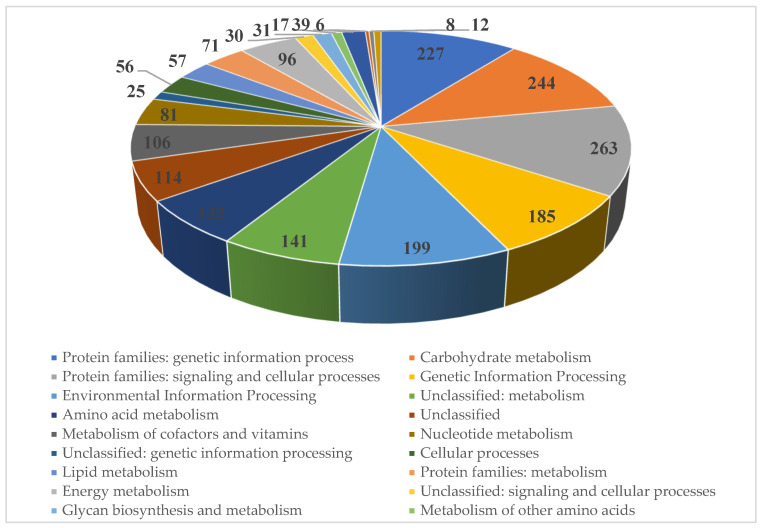
KEGG function classification of the *Streptomyces carpaticus* strain SCPM-O-B-9993.

**Table 1 biology-13-00388-t001:** Determination of antiviral activity of the suspension of strain *S. carpaticus* SCPM-O-B-9993 in laboratory conditions on tomatoes inoculated with CMV (each experimental group included ten plants).

Incubation Time, h	Number of Plants without Symptoms of Infestation					
	*S. carpaticus* SCPM-O-B-9993		K^+^		K^−^	
	pcs.	%	pcs.	%	pcs.	%
24	0	0	0	-	0	-
48	0	0	0	-	0	-
72	10	100	0	-	0	-
96	9	90	4	40	0	0
120	8	80	0	-	0	-
144	6	60	0	-	0	-
168	5	50	0	-	0	-

K^+^ positive control, plants treated with pharmaiodine 10%; K^−^ negative control, plants treated only with distilled water.

**Table 2 biology-13-00388-t002:** Genome parameters of strain SCPM-O-B-9993 and its closest relatives.

	Genome Size, bp	GC Content, %
*Streptomyces carpaticus* SCPM-O-B-9993_CP104005.1	5,968,715	72.84
*Streptomyces harbinensis* NA02264_CP054938.1	5,802,668	72.89
*Streptomyces xiamenensis* 318_CP009922.3	5,961,402	72.02
*Streptomyces* sp. XC2026_CP064057.1	5,836,896	72.10

**Table 3 biology-13-00388-t003:** The ANI and DDH values for strain SCPM-O-B-9993 and its closest relatives.

	ANI Value, %	DDH Value, %
*Streptomyces harbinensis* NA02264	98.71	90.90
*Streptomyces xiamenensis* 318	87.05	60.50
*Streptomyces* sp. XC 2026	86.87	58.00

**Table 4 biology-13-00388-t004:** The ANI and DDH values for strains SCPM-O-B-9993 and *Streptomyces harbinensis* NA02264 compared to those of the type strain of *S. harbinensis*.

	ANI Value, %	DDH Value, %
*Streptomyces harbinensis* NA02264	99.03	96.20
*Streptomyces carpaticus* SCPM-O-B-9993	98.72	90.40

**Table 5 biology-13-00388-t005:** Production clusters of secondary metabolites in strain SCPM-O-B-9993.

Type	Most Similar Known Cluster	Position, from	Position, to	Cluster Length, bp	Similarity with the Most Similar Known Cluster, %
NRPS	ohmyungsamycin A/ohmyungsamycin B	345,372	455,838	110,466	93
T1PKS	pellasoren	526,710	571,823	45,113	83
ectoine	ectoine	1,213,541	1,223,945	10,404	100
NRPS	coelibactin	5,391,062	5,451,702	60,640	81
T3PKS	naringenin	5,831,966	5,873,087	41,121	100
NAPAA, terpene	ε-poly-L-lysine	5,878,144	5,925,132	46,988	100

NRPS—non-ribosomal peptide synthetase; T1PKS—Type I PKS (Polyketide synthase); T3PKS—Type III PKS; NAPAA—non-alpha poly-amino acids like ε-poly-L-lysine.

## Data Availability

The data on genome sequence of the strain *S. carpaticus* strain SCPM-O-B-9993 can be found in the Genbank database under the accession number CP104005.1 (BioProject PRJNA269675, BioSample SAMN30493425).

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
