# Peer review of "Whole Genome Analysis and Assessment of the Metabolic Potential of Streptomyces carpaticus SCPM-O-B-9993, a Promising Phytostimulant and Antiviral Agent"

_biology, 2024, doi:10.3390/biology13060388_

Round 1

Reviewer 1 Report

Comments and Suggestions for Authors

Bataeva et al. obtained a promising plant-protecting and plant-stimulating strain isolated from brown semi-desert soils with very high salinity and analyzed its genome. They found this strain possibly produced antimicrobial properties (ohmyungsamycin, pellasoren, naringenin, and ansamycin) based on cluster analysis, and further investigated its antiviral activity against cucumber mosaic virus affecting tomatoes under laboratory conditions. Despite Streptomyces carpaticus SCPM-O-B-9993 might be a promising microorganism, the current manuscript lacks novelty and solid experimental data to confirm the capability of this strain, which could not fulfill the publication standards of Biology.

Usually, we first identify the valuable biological functions of a strain before analyzing its genome and recommending potential active metabolites for validation. However, the author speculated based on gene cluster analysis that the strain could produce bioactive compounds and then tested the antiviral activity of the strain. Firstly, the author did not confirm whether the strain actually produces these antiviral compounds because many gene clusters in Streptomyces are silent. Secondly, the author should provide more data to demonstrate the antiviral effectiveness of the strain, such as pot experiments, etc. Additionally, the resolution of Figures are too low.

Comments on the Quality of English Language

The language should be further improved. 

Author Response

We would like to thank the reviewer for valuable comments.

All the corrections made to the manuscript were highlighted in color.

We have improved English.

Indeed, we were initially determining properties of the strain. Therefore, following your recommendation, we have moved the sections (description of properties and genome analysis) to each other’s places.

We have conducted different experiments that proved properties of the strain and the composition of metabolites, which had been reported in the publications listed earlier:

(Biological activity and composition of metabolites of potential agricultural application from Streptomyces carpaticus K-11 RCAM04697 (SCPM-o-b-9993) / Yu. V. Bataeva, L. N. Grigoryan, A. G. Bogun [et al.] // Microbiology. – 2023. – Vol. 92, No. 3. – P. 459-467. – DOI 10.1134/s0026261723600155. – EDN OSXCII.) and

(Study of metabolites of Streptomyces carpaticus RCAM04697 for the creation of environmentally friendly plant protection products/ Yu. V. Bataeva, L. N. Grigoryan, E. A. Kurashov [et al.] // Theoretical and Applied Ecology. – 2021. – No. 3. – P. 172-178. – DOI 10.25750/1995-4301-2021-3-172-178. – EDN PUEIBM.)

In this paper, we have attempted to focus on whole-genome analysis. We were unable to prove whether the strain synthesizes antiviral compounds. Isolating the compounds and investigating their properties will be the next stage of our study.

Reviewer 2 Report

Comments and Suggestions for Authors

The article presents a comprehensive analysis of the whole genome and metabolic potential of Streptomyces carpaticus SCPM-O-B-9993, highlighting its role as a phytostimulant and antiviral agent by Evaluating phytostimulatory and antiviral properties of SCPM-O-B-9993 exhibited antiviral activity against cucumber mosaic virus affecting tomatoes under laboratory conditions. However, the written report lacks clarity and logical organization in describing the results. Therefore, addressing these concerns is crucial before considering it for acceptance.

Here are some suggestions for potential revisions or improvements:

1.     The abstract should include a concise summary of the conclusions and significance of this study to enhance comprehension.

2.     Re-organize the content in a logical flow. In introduction, ensure that the introduction clearly outlines the significance of studying Streptomyces carpaticus SCPM-O-B-9993, its potential applications as a phytostimulant and antiviral agent, and the objectives of the study, moving additional information to results, and/or discussion.

3.     The majority of this article lacks a definitive conclusion. For instance, sentences 53-59 merely present the BGCs of S. clavuligerus, S. tendae, and S. avermitilis without establishing their relevance to the Introduction section. Similarly, sentences 202-208 describe the biosynthetic pathways of precursors for secondary metabolites in SCPM-O-B-9993 as an indication of its metabolic potential; however, a conclusive statement is needed.

4.     The complete names of ANI and DDH should be provided upon their initial mention.

5.     In Materials and Methods, the authors should provide more detailed information on the methods used for genome analysis, identification of secondary metabolites, and assessment of antiviral activity. This will help readers understand the study's scientific rigor.

6.     In results 3.3 and 3.4, please provide a clear presentation and interpretation of the study findings, specifically focusing on the identification of gene clusters responsible for encoding secondary metabolites and elucidating the antiviral activity exhibited by the strain. Furthermore, discuss the significance of these discoveries within the realm of plant protection and stimulation, while also delving into the implications that Streptomyces carpaticus SCPM-O-B-9993's metabolic potential holds for agriculture and plant health. Lastly, explore practical applications for utilizing these identified secondary metabolites.

7.     Table 4, please unify the name of ε-Poly-L-lysine and e-polylysin.

Comments on the Quality of English Language

The manuscript needs to be completely reorganized.

Author Response

We would like to thank the reviewer for valuable comments.

All the corrections made to the manuscript were highlighted in color.

1. The abstract should include a concise summary of the conclusions and significance of this study to enhance comprehension.

This work aimed to study the genome organization and the metabolic potential of Streptomyces carpaticus strain SCPM-O-B-9993, a promising plant-protecting and plant-stimulating strain isolated from brown semi-desert soils with very high salinity. The strain genome contains a linear chromosome 5,968,715 bp long and has no plasmids. The genome contains 5,331 coding sequences among which 2,139 (40.1%) are functionally annotated. Biosynthetic gene clusters of secondary metabolites exhibiting antimicrobial properties (ohmyungsamycin, pellasoren, naringenin, and ansamycin) were identified in the genome. The most efficient period of SCPM-O-B-9993 strain cultivation was 72 hrs: during this period, the culture went from the exponential to the stationary growth phase as well as exhibited excellent phytostimulatory properties and antiviral activity against the cucumber mosaic virus in tomatoes under laboratory conditions. The Streptomyces carpaticus SCPM-OB-9993 strain is a biotechnologically promising producer of secondary metabolites exhibiting antiviral and phytostimulatory properties.

  1. Re-organize the content in a logical flow. In introduction, ensure that the introduction clearly outlines the significance of studying Streptomyces carpaticusSCPM-O-B-9993, its potential applications as a phytostimulant and antiviral agent, and the objectives of the study, moving additional information to results, and/or discussion.

We have made every effort to make respective corrections.

  1. The majority of this article lacks a definitive conclusion. For instance, sentences 53-59 merely present the BGCs of S. clavuligerus, S. tendae, and S. avermitilis without establishing their relevance to the Introduction section. Similarly, sentences 202-208 describe the biosynthetic pathways of precursors for secondary metabolites in SCPM-O-B-9993 as an indication of its metabolic potential; however, a conclusive statement is needed.

Corrections have been made (shown in color).

  1. The complete names of ANI and DDH should be provided upon their initial mention.

The following names have been provided:

the average nucleotide identity (ANI)….

the DNA-DNA hybridization (DDH) value…

  1. In Materials and Methods, the authors should provide more detailed information on the methods used for genome analysis, identification of secondary metabolites, and assessment of antiviral activity. This will help readers understand the study's scientific rigor.

The Materials and Methods section has been completely rewritten.

  1. In results 3.3 and 3.4, please provide a clear presentation and interpretation of the study findings, specifically focusing on the identification of gene clusters responsible for encoding secondary metabolites and elucidating the antiviral activity exhibited by the strain. Furthermore, discuss the significance of these discoveries within the realm of plant protection and stimulation, while also delving into the implications that Streptomyces carpaticus SCPM-O-B-9993's metabolic potential holds for agriculture and plant health. Lastly, explore practical applications for utilizing these identified secondary metabolites.

We have made every effort to make respective corrections.

  1. Table 4, please unify the name of ε-Poly-L-lysine and e-polylysin.

We have changed the name to ε-poly-L-lysine

Reviewer 3 Report

Comments and Suggestions for Authors

The content is clear. I suggest very few minor corrections:

i) Add an additional and more recent review to complement current reference 3 (which is a bit outdated)

ii) Use BGC (biosynthetic gene cluster) rather than BAC.

iii) Resolution of Fig. 2 is extremely poor. Please use a version of the figure with much better resolution.

Comments on the Quality of English Language

The last paragraph "Genome analysis is a promising field emphasizing the microbial biosynthesis pathways and facilitating the search for active metabolites. Currently, there is no alternative to genome analysis for rapid searching for new metabolites, including terpenes and antibiotics. Therefore, Streptomyces carpaticus strain SCPM-O-B-9993 is a biotechnologically promising producer of secondary metabolites with important properties."  must be rewritten

There are alternatives to genome analysis, such as metabolomic analysis (coupled to dereplication of known compounds) to assess the presence of potentially new compounds in any Streptomyces culture. The sentence "Currently, there is no alternative to genome analysis for rapid searching for new metabolites, including terpenes and antibiotics." can be eliminated and conclude with the current sentence by indicating the propterties of interest for the context of the paper (phytoprotection, etc.)

Author Response

We would like to thank the reviewer for valuable comments.

  1. Add an additional and more recent review to complement current reference 3 (which is a bit outdated)

All the corrections made to the manuscript were highlighted in color.

The link has been replaced. Review revised.

  1. ii) Use BGC (biosynthetic gene cluster) rather than BAC.

replaced

iii) Resolution of Fig. 2 is extremely poor. Please use a version of the figure with much better resolution.

Made a much better resolution version of the figure.

Comments on the Quality of English Language

The last paragraph "Genome analysis is a promising field emphasizing the microbial biosynthesis pathways and facilitating the search for active metabolites. Currently, there is no alternative to genome analysis for rapid searching for new metabolites, including terpenes and antibiotics. Therefore, Streptomyces carpaticus strain SCPM-O-B-9993 is a biotechnologically promising producer of secondary metabolites with important properties."  must be rewritten

There are alternatives to genome analysis, such as metabolomic analysis (coupled to dereplication of known compounds) to assess the presence of potentially new compounds in any Streptomyces culture. The sentence "Currently, there is no alternative to genome analysis for rapid searching for new metabolites, including terpenes and antibiotics." can be eliminated and conclude with the current sentence by indicating the propterties of interest for the context of the paper (phytoprotection, etc.)

Agree with the comments. Part of the text has been rewritten according to your recommendation.

We have improved English.

Reviewer 4 Report

Comments and Suggestions for Authors

Overall, I liked the article very much. It is easy to follow since it is direct, clear and concise.

The authors comprehensively presented the species/strain they worked with. Still, I also feel that the short presentation they provided on the systematics/taxonomy of the microorganism in question was somehow confusing and almost dependent on the sequencing of the specimen type of the species. If that was so important, why didn't they?

I would like to see the evidence showing that there are no plasmids present in the strain of Streptomyces carpaticus they've analysed.

Author Response

We would like to thank the reviewer for valuable comments.

The whole-genome analysis was described in the publication:

Biological activity and composition of metabolites of potential agricultural application from Streptomyces carpaticus K-11 RCAM04697 (SCPM-o-b-9993) / Yu. V. Bataeva, L. N. Grigoryan, A. G. Bogun [et al.] // Microbiology. – 2023. – Vol. 92, No. 3. – P. 459-467. – DOI 10.1134/s0026261723600155. – EDN OSXCII.

Genomic DNA was isolated by phenol–chloroform extraction. Whole-genome sequencing was conducted on the Illumina MiSeq platform using Nextera DNA Library Preparation Kit and MiSeq Reagent Kit v3. Monomolecular nanopore DNA sequencing was performed on the MinION platform in compliance with the manufacturer’s recommendations using a rapid barcoding kit (RBK004) and a MinION flow cell (R9.4.1). Sequencing was carried out using the MinKNOW v18.05.5 software (time, 48 hrs; 180 mV); demultiplexing, using the Guppy v6.0.1 software. Hybrid genome assembly without preliminary read trimming was performed using the Unicycler v0.4.7 software.

Whole-genome sequencing on the Illumina MiSeq and minION platforms yielded 916,371 short reads (533,148,207 bp) and 103,371 short reads (247,666,704 bp), respectively. The whole-genome sequence of the Streptomyces carpaticus strain was deposited to the NCBI GenBank database for the first time under accession number CP104005.1. Annotation was performed using NCBI Prokaryotic Genome Annotation Pipeline (PGAP) GeneMarkS-2+ (revision 6.2). The final assembled genome consisted of one linear chromosome 5,968,715 bp long. The GC composition, 72.84%. A total of 5206 protein-coding sequences, 60 tRNA sequences, 15 rRNA sequences (5 – 5S, 5 – 16S, 5 – 23S), and eight CRISPR loci were identified during genome annotation and analysis.

The whole-genome sequencing software demonstrated that it was a plasmid-free linear chromosome. We did not conduct any specialized plasmid detection. Confirmation is probably needed.

Round 2

Reviewer 1 Report

Comments and Suggestions for Authors

The authors have improved the manuscript. although they declared they have conducted different experiments that proved properties of the strain and the composition, while most of them were sequencing data analyses rather than experimental data. Despite genome analysis is important, while basic analyses is insufficient.

Reviewer 2 Report

Comments and Suggestions for Authors

The authors appear to have addressed all the issues I was concerned about.